# Why do ambulance employees (not) seek organisational help for mental health support? A mixed-methods systematic review protocol of organisational support available and barriers/facilitators to uptake

Sasha Johnston ,[1,2] Kristy Sanderson ,[3] Lucy Bowes,[2] Jennifer Wild [2,4]

[1]South Western Ambulance Service NHS Foundation Trust, Bristol, UK
[2]Department of Experimental Psychology, University of Oxford, Oxford, UK
[3]School of Health Sciences, University of East Anglia, Norwich, United Kingdom, Norwich, UK
[4]Oxford Health NIHR Biomedical Research Centre, Oxford, UK

**Correspondence to**
Sasha Johnston;
sasha.johnston@psy.ox.ac.uk

## ABSTRACT

**Introduction** The COVID-19 pandemic is exacerbating a wide range of symptoms of poor mental health among emergency medical service (EMS) ambulance populations. Evidence suggests that using organisational support can improve employee outcomes and in turn, patient outcomes. Understanding why EMS staff do and do not use support services is therefore critical to improving uptake, ensuring equitable access, and potentially influencing workforce well-being, organisational sustainability and patient care delivery. This systematic review aims to identify what support is available and any perceived barriers and facilitators to accessing and utilising organisational support.

**Methods and analysis** Searches performed between 18 February 2022 and 23 February 2022 will be used to identify studies that report barriers and facilitators to EMS employee support among all government/state commissioned EMS ambulance systems. Electronic databases, AMED, CINAHL, Cochrane Central Register of Controlled Trials, Cochrane Database of Systematic Reviews, EMBASE, EMCARE, HMIC, Medline and PsycINFO will be searched. All relevant English-language studies of adult employees of government/state commissioned EMS ambulance organisations published since December 2004 will be screened and relevant data extracted by two independent reviewers. A third reviewer will resolve any disagreements.

The primary outcome is the identification of perceived barriers or facilitators to EMS staff using organisational support for mental health. The secondary outcome is the identification of supportive interventions offered through or by ambulance trusts. Study selection will follow Preferred Reporting Items for Systematic Reviews and Meta-Analyses guidelines, and the methodological quality of included studies will be appraised by administering rating checklists. A narrative synthesis will be conducted to report qualitative and quantitative data and will include population characteristics, methodological approach and information about barriers and facilitators.

**Ethics and dissemination** Ethical approval is not required because only available published data will be analysed. Findings will be disseminated through peer-reviewed publication and conference presentation.

**PROSPERO registration number** CRD42022299650.

## STRENGTHS AND LIMITATIONS OF THIS STUDY

⇒ This systematic review addresses a gap in the current evidence-base by providing an overview and critical appraisal of studies that report emergency medical service employee perceptions of the barriers and facilitators to organisational mental health support, which may influence employee uptake of such support.
⇒ By following the Preferred Reporting Items for Systematic Review and Meta-Analysis Protocols and Synthesis without meta-analysis in systematic reviews reporting guidelines and by registering and publishing this protocol, the transparency of systematic review methods and findings is improved.
⇒ Restricting the study to English-language only publications may exclude relevant information written in other languages.
⇒ There is potential for heterogeneous and low-quality reporting of barriers and facilitators in the studies identified for review.

## INTRODUCTION

Emergency medical service (EMS) employees save lives. They respond to emergency and urgent care needs to reduce anxiety, pain, and suffering. EMS is called to work in a range of environments and with a range of patient populations, undertaking autonomous life and death decisions. They frequently work long, irregular hours, while contending with staff shortages and exposure to distressing and traumatic events. These factors can result in severe consequences for some staff; with an increased risk of post-traumatic stress, early retirement on medical grounds, accidental injury or death.[1] EMS employees are over four times more likely to experience mental ill health compared with the general workforce.[2 3] The COVID-19 pandemic further exacerbates risk of poor mental health. A recent survey of UK emergency responders

identified that ambulance staff (77%) were the most likely to report their mental health has worsened since the pandemic began.[4] Suicide is a particular concern,[5] with Mars *et al*[6] identifying a 75% increased risk among male paramedics compared with the general population.

A number of risk factors contribute to EMS employee mental ill health, including those shared with the general population, such as genetics, loneliness, stressful life events and physical ill health.[7] A recent systematic review identified a higher prevalence of alcohol and drug misuse compared with the general population[3] and evidence suggests a high prevalence of adverse childhood experiences among EMS employees, such as abuse and neglect.[8] However, research by the mental health charity Mind[9] found that EMS employees were twice as likely as the general population to identify problems at work as the main cause of their mental ill health. Poor employee mental health can have a detrimental impact on EMS capability, with some areas reporting a 50% staff attrition rate, citing poor staff mental health and organisational culture cited as primary contributing factors.[10 11] Evidence suggests that utilising organisational support when needed is related to improved employee and patient outcomes.[12] If support is not available or employees are unable or won't access support, staff may feel isolated, unsupported and this can lead to poor mental health and an inability to thrive at work.[13] EMS employees report reluctance to disclose mental health problems at work, citing perceived stigma associated with mental health and feeling unsupported by employers to address mental well-being. To help prevent workforce burnout, action is needed to better support EMS employee mental well-being. With the right support, staff experiencing mental ill health can successfully continue to work, the severity of symptoms can be reduced and suicide prevented.[14 15] In addition the frequency and length of sickness absence reduces; increasing workforce productivity, capability and safety.[16] Current EMS employee assistance programme uptake is improving, but it is vital that EMS organisations make improvements to ensure all employees can access support when needed.[17] This is vital not only because of the impact of poor mental health on individual employees, but also the critical impact of prehospital care on patient outcomes.[18] Therefore, understanding what EMS employees perceive to be barriers and facilitators to utilising support services is key to improving their uptake. This systematic review aims to improve our understanding of why some employees access organisational support and why others do not. This protocol aims to provide a transparent method of identifying current support provision, barriers and facilitators to utilising support, while assessing the quality and risk of bias of the current available evidence.

## Review aim

Our primary aim is to identify and review previously conducted studies which include reports of EMS ambulance employees' perceptions of the barriers or facilitators to the provision of organisational mental health support for their own psychological well-being.

## Objectives

The objective is to establish what support is available and identify any element/s perceived as effective and/or ineffective for the uptake of organisational support for EMS employee mental well-being. For the purposes of this review organisational support is defined as any programme, pathway or signposting that is provided, funded or facilitated by the employing organisation in support of mental health. This review will seek to:

► Identify and report the range of the distinct types of supportive interventions available for EMS ambulance employee well-being.
► Establish the proportions of participants that report barriers and/or facilitators and/or other key factors.
► Identify attitudes, perceptions and experiences relating to any barriers, facilitators and other key factors.

## METHODS AND ANALYSIS

This protocol was prepared following the Preferred Reporting Items for Systematic Review and Meta-Analysis Protocols (PRISMA) and Synthesis without meta-analysis (SWiM) in systematic reviews reporting guidelines.[19 20] The protocol was then registered with the International Prospective Register of Systematic Reviews (PROSPERO) on 2 February 2022 (ref CRD42022299650).

## Inclusion and exclusion criteria
### Types of studies

All study types that examine factors relating to organisational mental health support for prehospital EMS ambulance organisation employees will be included. Primary papers from relevant systematic reviews alongside quantitative, and mixed-methods studies will be included to establish what interventions are offered and to assess barriers, facilitators, and any associated benefits and/or harms linked to reported interventions. Qualitative, cross-sectional and survey studies that report any barriers and/or facilitators relating to organisational employee mental health support, will also be examined.

Only articles published after 1 December 2004 will be examined, since this date coincides with a shift in focus on the well-being of first responders across the globe. This shift likely relates to the terrorist attack in New York on 11 September 2001. Legislative and guidance changes were introduced to ambulance organisations across the globe such as 'Agenda for change' (2004)[21] in the UK, in the USA, the 'EMS Workforce for the 21st Century project'[22] commenced in the fall of 2004 and in Australia the 'Emergencies Act (ACT)' (2004)[23] promoted responder welfare and described employer responsibility. Articles not written in English will be excluded. Any study samples that consist of mixed emergency employees (ambulance/coastguard/fire/police), where results are combined,

and samples include less than 50% ambulance staff will also be excluded from this systematic review.

## Types of participants

All studies involving adults (18+) employed by government or state commissioned EMS organisations in clinical or non-clinical roles will be included. Employees will be eligible for inclusion if contracted to full or part-time roles or hold a bank contract that requires a minimum number of regular working hours. Employees could include paramedics, Emergency Medical Technicians, Emergency Care Assistants, EMS ambulance nurses and doctors, emergency medical number call centre and dispatch staff, operational managers, support and central function staff such as Human Resources and patient safety teams, as well as senior leadership. Paramedic students, EMS apprentices, non-government/state commissioned/ private EMS ambulance employees and volunteers, including volunteer first responders will be excluded, since any available supportive interventions may differ from those offered to employed staff.

## Interventions

The review will include studies which report on EMS ambulance employees' perceived barriers or facilitators to seeking or accessing help from their organisation for mental health support. This may include individual-level factors relating to the decision to engage in employee support, the acceptability of the support offered, perceptions and experiences of support, as well as organisational level factors such as, culture, and finally, policy level factors such as targeted campaigns and regulation of professional standards. Organisational factors examined in this review will include interventions reported to be offered for employee mental health and well-being. Any intervention, regardless of the mode of delivery (face to face, e-learning, virtual, etc), is eligible for inclusion if the employer was involved in any element such as development, design, delivery, funding, signposting. Studies that only examine social support (support outside of the employee context, such as non-organisational family and friend support) and organisational support in response to isolated specialist occurrences, such as natural disaster and terrorist events, will be excluded.

The main outcome will be the identification of EMS ambulance employees' perceived barriers or facilitators to accessing organisational support for their mental health (including formal peer-support networks, manager support and employee assistance programmes). This will Include elements of organisational factors identified by participants as being effective or ineffective for the provision and uptake of support. The presence of any factor that promotes the development, implementation, adoption, uptake of or participation with, organisational employee mental health support will be considered a facilitator. Any factor that limits or restricts the development, implementation, adoption, uptake of or participation with organisational employee mental health support will be considered as a barrier. The same factor may be both a barrier and a facilitator.

## Information sources

The following electronic databases were searched between 18 February 2022 and 23 February 2022 (and will be rerun 6 weeks before review completion): AMED, CINAHL, Cochrane Central Register of Controlled Trials and the Cochrane Database of Systematic Reviews via the Cochrane Library, EMBASE, EMCARE, HMIC, Medline, PsycINFO, Scopus and Web of Science . An example search strategy for Medline is presented in online supplemental appendix 1. Searches were tailored to each database using the Polyglot Search Translator[24] and conducted using keywords and relevant theasai such as MeSH and EMTREE. To ensure that all the available and relevant research is captured, grey literature will also be sought from the OpenGrey, MedNar and ProQuest databases and through the webpages of industry and charitable organisations active in supporting EMS ambulance employee mental health. A full list of webpages to be manually searched will be developed by the research team and will include sites such as the Global Ambulance Leadership Alliance (which covers the UK, USA, Canada and Australasia), The Ambulance Staff Charity (UK), the Royal Foundation, and the mental health charity, Mind. The reference lists of all studies selected for critical appraisal will be hand searched for further material for inclusion. The searches will be rerun 6 weeks prior to the final analyses to identify and retrieve any other studies for inclusion.

## Study records
### Data management

References identified from electronic and hand searches, including title and abstracts, will be imported into Mendeley citation manager software and any duplicates removed.

### Selection process

Two reviewers will independently screen a subset (10%) of titles and abstracts. Full-text screening will be based on a PICoT concept:

► Population: Adults (18+) employed by government/ state commissioned EMS ambulance services.
► Phenomena of Interest: Types of organisational interventions offered to support ambulance staff mental health and any barriers and/or facilitators to utilising such support.
► Context: Government/state commissioned prehospital EMS ambulance organisations.
► Types of studies study design: All types of research studies.

Studies scoring 4/4 for all the above criteria will be included. Any reviewer uncertainty will be rated as 'unsure' and discussed by the independent reviewers with reference to the full text if required. If not resolved through reviewer discussion, disagreements will be settled

through discussion with an independent third reviewer. The inter-rater reliability of consensus will be calculated.

## Data extraction process

To identify papers for inclusion the full text of remaining studies will be retrieved and screened. Again, the inter-rater reliability will be calculated to ensure consistency and clarity. From this final selection, all potentially relevant data will be extracted and collated in an Excel spreadsheet including:

► Primary author.
► Publication details.
► Country of study.
► Study methods.
► Setting.
► Sample characteristics (sample size, age range, EMS job role).
► Phenomenon of interest (self-reported barriers and/or facilitators).
► Intervention (where relevant).

   Outcomes measured will include:

► Primary outcome measures (self-reported barriers and/or facilitators).
► Assessment tool names.
► Reported statistics.
► Reported significance levels.
► Reported effect sizes.
► Secondary outcome measures.
► Relevant findings.

   To ensure sufficient detail capture to enable replication, any described intervention content will be extracted using Hoffman et al's[25] Template for Intervention Description and Replication checklist. If data are missing or additional information is required, we will contact authors by email as per Cochrane recommendations and document the frequency of contact and authors' responses.[26] Search results will be reported in full and presented in a PRISMA flow diagram.

## Quality assessment

The quality, alongside the trustworthiness, relevance and findings of each of the studies identified for final selection will be assessed by two independent reviewers using two rating checklists (Standard Quality Assessment Checklists) developed by Kmet et al.[27] One checklist is designed to assess the quality for quantitative studies (and will also be applied to the quantitative components of mixed-methods studies) and the other for qualitative studies (which will also be applied to the qualitative components of mixed-methods studies). Each checklist item will be rated on a quality scale from 0 to 2:

► Criteria not met=0.
► Criteria partially met=1.
► Criteria fully met=2.

   Any included grey literature will be assessed using Tyndall's[28] 'Authority, Accuracy, Coverage, Objectivity, Date, Significance' checklist. Reviewer discrepancies will be resolved through discussion and when necessary,

consultation with the third reviewer. All study types will be included in this review, regardless of methodological quality, since it is anticipated that the availability of high-quality evidence will be limited. However, a sensitivity analysis will be conducted by testing whether removing any studies rated zero for methodological quality from the analysis changes the thematic results. Critical appraisal results will be displayed in a predetermined assessment of methodological quality table. The narrative synthesis will include a summary of the relative impact of missing data and of methodological flaws on the findings.

## Data synthesis

Mixed-methods systematic reviews are an emerging field of enquiry, useful for enhancing the credibility of findings. This is particularly important for this review as although quantitative evidence suggests that ambulance staff report high rates of mental ill health and want organisational support,[9 29] evidence from qualitative studies indicates that negative experiences and perceptions of such support can affect the acceptability of utilising support.[30] By using a mixed-methods approach, both the experience and effectiveness of organisational support initiatives can be captured; factors vital for informing the research question. The mixed-methods procedure will follow Joanna Briggs Institute guidance for a convergent integrated approach.[31] This involves transforming extracted data from quantitative papers (and quantitative aspects of mixed-methods papers) by qualitising (creating a textual description) quantitative findings. This enables findings from all studies to then be combined during the analysis phase. It is anticipated that data from the included studies will be heterogeneous since they are likely to include different approaches to design and use of different outcome measures. Heterogeneity will be determined by summarising:

► Population characteristics (eg, sample size, age, type of mental health problem/disorder).
► Methodological approach (eg, qualitative, survey, experiment).
► Assessment (the measures used to assess staff perceptions of organisational support, barriers or facilitators where relevant).
► Intervention characteristics (eg, type of intervention, frequency, duration, uptake).

   It is therefore unlikely that it will be possible to undertake a meta-analysis. Instead, a narrative review and synthesis approach will be taken by conducting inductive thematic analysis, using NVivo software and data from the excel data extraction sheets in the following steps:

1. Key data (data from the results sections of included papers) and quotations will be transposed from data extraction sheets to NVivo for coding by two reviewers, who will agree a coding structure for coding of participant data. The third reviewer will arbitrate any conflict.
2. Using the agreed on coding structure, two reviewers will undertake thematic analysis of the coded data and will meet regularly to ensure the coding structure is ap-

propriate and can be applied to the conclusions being drawn from the identified themes.

3. Factors impacting on participation with EMS organisational employee mental health support, with a focus on information from studies relating to employee experiences and/or perceptions of barriers against, or facilitators to, accessing and using support, will be synthesised in this systematic review (there will be no minimum number of studies).

4. A combined narrative (descriptive) synthesis will be used following Campbell *et al*'s[20] SWiM guideline.

5. The certainty of evidence will also be synthesised using #27 quality checklist.

## Amendments

If any protocol amendments are required, the date, description and rationale will be made available on the PROSPERO registration.

## Patient and public involvement

To enhance the meaningfulness and robustness of findings, an EMS staff reference group and an EMS specific patient involvement group in the UK reviewed and supported the development and design of this protocol. These groups will review and provide an employee and public perspective on the interpretation of the findings and will support dissemination.

## Ethics and dissemination

Ethical approval is not required because only available published data will be analysed and this is a protocol for a systematic review. Findings will be disseminated through publication in a relevant peer-reviewed journal. The findings will also be communicated at research conferences, symposia, congresses and via social media to ensure dissemination to a wide range of interested parties.

## DISCUSSION

EMS employee mental well-being can influence the care given to patients. A number of initiatives are provided to support EMS employee mental health, although evidence suggests that some staff do not seek help or feel unable to disclose their mental health status when needed. With this in mind, a strength of this systematic review will be the presentation of barriers and facilitators specific to the uptake of employee mental health support in the EMS context identified through robust, replicable methods and critical appraisal of the available literature. Limitations will be addressed through transparent reporting and appraisal of study quality the involvement of EMS staff in the development of the inclusion and exclusion criteria, and by grading of the quality of the studies included.

**Acknowledgements** We acknowledge the help and support of senior librarians Laura Coysh, Plymouth Discovery Library, Derriford Hospital, Plymouth and Karine Barker, Radcliffe Science Library, Bodleian libraries, University of Oxford. We are grateful for the funding provided by NHS England and NHS Horizons and for the support of South Western Ambulance NHS Foundation Trust.

**Contributors** This study concept and design were conceived by authors SJ, JW and KS. SJ and JW drafted this manuscript with support from KS and LB who reviewed and edited the final version. All approved the final submission.

**Funding** This work was supported by NHS England and NHS Horizons 'Project A' and by South Western Ambulance NHS Foundation Trust as part of SJ's DPhil in Experimental Psychology with the University of Oxford. JW is supported by MQ (CQRO1260), the Wellcome Trust (00070) and Oxford Health NIHR Biomedical Research Centre.

**Competing interests** None declared.

**Patient and public involvement** Patients and/or the public were involved in the design, or conduct, or reporting, or dissemination plans of this research. Refer to the Methods section for further details.

**Patient consent for publication** Not applicable.

**Provenance and peer review** Not commissioned; externally peer reviewed.

**ORCID iDs**
Sasha Johnston http://orcid.org/0000-0002-9904-8834
Kristy Sanderson http://orcid.org/0000-0002-3132-2745
Jennifer Wild http://orcid.org/0000-0001-5463-1711

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
