## [Reviewer comments · BMJ Open]

ARTICLE DETAILS

TITLE (PROVISIONAL)	Why do ambulance employees (not) seek organisational help for mental health support? A mixed-methods systematic review protocol of organisational support available and barriers/facilitators to uptake
AUTHORS	Johnston, Sasha; Sanderson, Kristy; Bowes, Lucy; Wild, Jennifer

VERSION 1 – REVIEW

REVIEWER	Rowe , Sarah King's College London, Health Service and Population Research
REVIEW RETURNED	11-Apr-2022

GENERAL COMMENTS	Why do ambulance employees (not) seek organisational help for mental health support? A mixed-methods systematic review protocol This protocol has good potential, and it is an interesting topic. It was great to see you had thought about how to incorporate public patient involvement within your systematic review, as this is often overlooked. However, some areas in the protocol were inconsistent and unclear. It would be very helpful to have the following areas addressed: Major Comments: • I don't really understand what is meant by a 'mixed-methods' systematic review. What is this? Is it simply a review that includes qualitative and quantitative papers or is it one where you are doing a meta-synthesis of the qualitative papers and a meta-analysis for the quantitative papers? This needs some explanation as to what it is and rationale for why this method is being chosen.• It states under the objectives (page 5, line 53), that interventions available for EMS ambulance employee wellbeing are being identified and reported in this review. It makes no mention of this in the title, abstract or aims. This is a missed opportunity to highlight the further usefulness of this review. It is also an important detail as it means there are two distinct review questions that are the focus of this paper. It would be helpful to clarify this point throughout the manuscript as it is inconsistent.• Types of studies to be included (page 6, line 35). Rather than including systematic reviews, you should ideally include any relevant primary (original) papers from these. It would be appropriate to check the reference lists of any relevant systematic reviews for eligible papers.• Page 10, the selection criteria follow PICOT and studies including all 4 criteria will be included. However, they do not list 'interventions' as one of the 'phenomena of interest' despite it being one of the
---

	review objectives. This should be added (see the first point about being consistent throughout).  • Some of the phrasing could be changed so it is more accurate as it was confusing/misleading to read in a few places. On page 11 line 45 and 50 refers to 'risk of bias' however, it is a quality assessment that is being conducted. These are different (https://handbook-5-1.cochrane.org/chapter_8/8_2_2_risk_of_bias_and_quality.htm) and the terminology should reflect that accurately. On page 11, it refers to 'heterogeneity' and a 'sensitivity analysis' being conducted but this implies there's a meta-analysis being conducted when there's not. It's unclear if you're removing the studies rated zero for quality from your analysis to see the impact on the results or are those considered 'low quality' excluded entirely from the review i.e. this is a part of your inclusion/exclusion criteria? • The analysis techniques in the systematic review are very confusing. Perhaps you need to have two subheadings which separate how you plan to analyse the results from quantitative papers from how you plan to analyse the results from qualitative papers. Are only qualitative papers included in the meta-synthesis? Are quantitative papers analysed using narrative synthesis? On page 12, line 36 – what is considered "Key data" that will be coded for the thematic analysis? Is this data in the results section of qualitative papers? More transparency and explanation are needed here. Minor comments:  • Page 9 lines 3-9 are repetitive and have been sufficiently covered earlier in the methods. This could be removed. • Page 10, line 35. It's stated that a subset (300) of titles and abstracts will be independently reviewed. Usually a percentage (e.g. 10%) of these are done rather than a specific number. • Is there a word missing on page 11, line 51-52 "The narrative will include...". Should this be "The narrative synthesis will include...?"
--	---

REVIEWER	Forson, Paa Kobina Komfo Anokye Teaching Hospital, Emergency Medicine
REVIEW RETURNED	05-May-2022

GENERAL COMMENTS	Dear authors, thank you very much for submitting this article. I think the presentation of the research protocol is very important. My thoughts on the protocol that it looks good. I am struggling to align your review aim with the objectives of this study. The overarching aim of the review appears similar to the third objective in terms of the characters of interest. It may be useful to clearly define the objectives with the review aim in mind - this is unless the study protocol requires that these are separately listed. Additionally, if you will be contacting authors who do not provide sufficient data in their article, would you use a structured documented consenting system.
---

VERSION 1 – AUTHOR RESPONSE

Reviewer: 1 Dr. Sarah Rowe , King's College London

Major Comments:

R1.1: I don't really understand what is meant by a 'mixed-methods' systematic review. What is this? Is it simply a review that includes qualitative and quantitative papers or is it one where you are doing a meta-synthesis of the qualitative papers and a meta-analysis for the quantitative papers? This needs some explanation as to what it is and rationale for why this method is being chosen.

> Thank you, we apologise that the terminology is unclear. We provide further detail and references to explain the rationale and the procedural method of the mixed-methods approach:

p.13 **Data synthesis**

Mixed-methods systematic reviews are an emerging field of enquiry, useful for enhancing the credibility of findings. **This is particularly important for this review as although quantitative evidence suggests that ambulance staff report high rates of mental ill health and want organisational support,[9,29] evidence from qualitative studies indicates that negative experiences and perceptions of such support can affect the acceptability of utilising support.[30] By using a mixed-methods approach, both the experience and effectiveness of organisational support initiatives can be captured; factors vital for informing the research question. The mixed-methods procedure will follow Joanna Briggs Institute guidance for a convergent integrated approach.[31] This involves transforming extracted data from quantitative papers (and quantitative aspects of mixed-methods papers) by qualitzing (creating a textual description) quantitative findings.** This enables findings from all studies to then be combined during the analysis phase.

References:

- 29 Dropkin J, Moline J, Power P, *et al.* A qualitative study of health problems, risk factors, and prevention among Emergency Medical Service workers. *Work* 2015;**52**:935–51. Doi:10.3233/wor-152139
- 30 Johnston S, Wild J, Sanderson K, *et al.* Perceptions and experiences of mental health support for ambulance employees. *Journal of Paramedic Practice*. <https://doi.org/1012968/jpar2022147287> 2022;**14**:287–96. doi:10.12968/JPAR.2022.14.7.287
- 31 Lizarondo L, Stern C, Carrier J, *et al.* Chapter 8: Mixed Methods Systematic Reviews. In: *JBI Manual for Evidence Synthesis*. JBI 2020. doi:10.46658/JBIMES-20-09

R1.2: It states under the objectives (page 5, line 53), that interventions available for EMS ambulance employee wellbeing are being identified and reported in this review. It makes no mention of this in the title, abstract or aims. This is a missed opportunity to highlight the further usefulness of this review. It is also an important detail as it means there are two distinct review questions that are the focus of this paper. It would be helpful to clarify this point throughout the manuscript as it is inconsistent.

> Thank you for bringing this important point to our attention. We have updated the following:

p.1 Title

Why do ambulance employees (not) seek organisational help for mental health support? A mixed-methods systematic review protocol **of organisational support available and barriers/facilitators to uptake.**

p.2 Abstract

Introduction

This systematic review aims to identify **what support is available** and any perceived barriers and facilitators to accessing and utilising organisational support.

Methods and analysis

The primary outcome of this systematic review is the **identification of** perceived barriers or facilitators to EMS staff utilising organisational support for mental health. The secondary outcome is the identification of **what supportive interventions** offered through or by ambulance Trusts.

p.6

Introduction

This protocol aims to provide a transparent method of **identifying current support provision**, barriers and facilitators to utilising support, whilst assessing the quality and risk of bias of the current available evidence.

Objectives

Our objective is to **establish what support is available** and identify any element/s perceived as effective and/or ineffective for the uptake **and delivery** of organisational support for EMS employee mental wellbeing

R1.3: Types of studies to be included (page 6, line 35). Rather than including systematic reviews, you should ideally include any relevant primary (original) papers from these. It would be appropriate to check the reference lists of any relevant systematic reviews for eligible papers.

>Thank you, paragraph amended accordingly

P7. Types of studies

Primary papers from relevant systematic reviews alongside quantitative, and mixed-methods studies will be included to **establish what interventions are offered and to** assess barriers, facilitators, and any associated benefits and/or harms linked to reported interventions.

R1.4: Page 10, the selection criteria follow PICOT and studies including all 4 criteria will be included. However, they do not list 'interventions' as one of the 'phenomena of interest' despite it being one of the review objectives. This should be added (see the first point about being consistent throughout).

>Thank you, p11 PICoT updated to read:

- **phenomena of Interest: Types of organisational interventions offered to support ambulance staff mental health** and any barriers and/or facilitators to utilising such support.

R1.5: Some of the phrasing could be changed so it is more accurate as it was confusing/misleading to read in a few places. On page 11 line 45 and 50 refers to 'risk of bias' however, it is a quality assessment that is being conducted. These are different (https://handbook-5-1.cochrane.org/chapter_8/8_2_2_risk_of_bias_and_quality.htm) and the terminology should reflect that accurately. On page 11, it refers to 'heterogeneity' and a 'sensitivity analysis' being conducted but this implies there's a meta-analysis being conducted when there's not. It's unclear if you're removing the studies rated zero for quality from your analysis to see the impact on the results or are those considered 'low quality' excluded entirely from the review i.e. this is a part of your inclusion/exclusion criteria?

>Thank you for your suggestions. On p12 we have clarified and updated the **Quality assessment** as follows:

All study types will be included **in this review**, regardless of **methodological quality**, since it is anticipated that the availability of high-quality evidence will be limited. However, a sensitivity analysis will be conducted **by testing whether removing any studies rated zero for methodological quality from the analysis changes the thematic results**. Critical appraisal results will be displayed in a pre-determined **assessment of methodological quality** table.

R1.6: The analysis techniques in the systematic review are very confusing. Perhaps you need to have two subheadings which separate how you plan to analyse the results from quantitative papers from how you plan to analyse the results from qualitative papers. Are only qualitative papers included in the meta-synthesis? Are quantitative papers analysed using narrative synthesis? On page 12, line 36 – what is considered “Key data” that will be coded for the thematic analysis? Is this data in the results section of qualitative papers? More transparency and explanation are needed here.

> Thank you. The analysis techniques have now been explained in further detail as per the response provided to Question R1.1. on p.13 ‘Data synthesis’

p.14 **Data synthesis** now updated to read:

Key data **(data from the results sections of included papers)** and quotations will be transposed from data extraction sheets to NVivo for coding by two reviewers, who will agree a coding structure for coding of participant data. The third reviewer will arbitrate any conflict.

Step 6. removed as repetitive

Minor comments:

- Page 9 lines 3-9 are repetitive and have been sufficiently covered earlier in the methods. This could be removed.

>Thank you, lines 3-9 removed

- Page 10, line 35. It’s stated that a subset (300) of titles and abstracts will be independently reviewed. Usually a percentage (e.g. 10%) of these are done rather than a specific number.

> Many thanks, p.11 amended to read 10%

- Is there a word missing on page 11, line 51-52 “The narrative will include...”. Should this be “The narrative synthesis will include...”?

> Yes, thank you for spotting this. We have re-written the sentence as suggested.

Reviewer: 2

Dr. Paa Kobina Forson, Komfo Anokye Teaching Hospital, St Patricks Hospital

Comments to the Author:

Dear authors, thank you very much for submitting this article. I think the presentation of the research protocol is very important.

My thoughts on the protocol that it looks good. I am struggling to align your review aim with the objectives of this study. The overarching aim of the review appears similar to the third objective in terms of the characters of interest.

It may be useful to clearly define the objectives with the review aim in mind - this is unless the study protocol requires that these are separately listed.

>Very many thanks, the review objectives have been refined following reviewer 1 comments and now align more clearly with the aim.

Additionally, if you will be contacting authors who do not provide sufficient data in their article, would you use a structured documented consenting system.

>Thank you for raising this query. We will follow Cochrane recommendations and will email the corresponding author for clarification or additional information. The frequency of contact and authors’ responses will be documented.

p12 ‘Data Extraction process’ sentence expanded and reference added:

If data are missing or additional information is required, **we will contact authors by email and as per Cochrane recommendations, and the frequency of contact and authors' responses will be documented.[26]**

Reference

26 Young T, Hopewell S. Methods for obtaining unpublished data. Cochrane. 2011. https://www.cochrane.org/MR000027/METHOD_methods-for-obtaining-unpublished-data (accessed 4 Aug 2022).

VERSION 2 – REVIEW

REVIEWER	Rowe , Sarah King's College London, Health Service and Population Research
REVIEW RETURNED	07-Sep-2022
GENERAL COMMENTS	The edits have improved the manuscript and it reads very well. There is one minor point that could be addressed: The sentence "(and will be re-run six weeks before review completion)" on page 9 line 53-54 is repeated on page 10 line 23-24. The sentence on page 9 could be deleted. After that edit, I would be happy to see this published and look forward to reading the completed systematic review.
REVIEWER	Forson, Paa Kobina Komfo Anokye Teaching Hospital, Emergency Medicine
REVIEW RETURNED	06-Sep-2022
GENERAL COMMENTS	Dear Author, congratulations for making the necessary changes to the manuscript